# Multi-Game Decision Transformers

**Kuang-Huei Lee**[*]     **Ofir Nachum**[*]     **Mengjiao Yang**     **Lisa Lee**

**Daniel Freeman**     **Winnie Xu**     **Sergio Guadarrama**     **Ian Fischer**

**Eric Jang**     **Henryk Michalewski**     **Igor Mordatch**[*]

Google Research

## Abstract

A longstanding goal of the field of AI is a method for learning a highly capable, generalist agent from diverse experience. In the subfields of vision and language, this was largely achieved by scaling up transformer-based models and training them on large, diverse datasets. Motivated by this progress, we investigate whether the same strategy can be used to produce generalist reinforcement learning agents. Specifically, we show that a single transformer-based model – with a single set of weights – trained purely offline can play a suite of up to 46 Atari games simultaneously at close-to-human performance. When trained and evaluated appropriately, we find that the same trends observed in language and vision hold, including scaling of performance with model size and rapid adaptation to new games via fine-tuning. We compare several approaches in this multi-game setting, such as online and offline RL methods and behavioral cloning, and find that our Multi-Game Decision Transformer models offer the best scalability and performance. We release the pre-trained models and code to encourage further research in this direction.[1]

## 1   Introduction

Building large-scale generalist models that solve many tasks by training on massive task-agnostic datasets has emerged as a dominant approach in natural language processing [18, 12], computer vision [19, 6], and their intersection [61, 4]. These models can adapt to new tasks (such as translation [63, 78]), make use of unrelated data (such as using high-resource language to improve translations of low-resource languages [17]), or even incorporate new modalities by projecting images into language space [46, 75]. The success of these methods largely derives from a combination of scalable model architectures [77], an abundance of unlabeled task-agnostic data, and continuous improvements in high performance computing infrastructure. Crucially, scaling laws [38, 31] indicate that performance gains due to scale have not yet reached a saturation point.

In this work, we argue that a similar progression is possible in the field of reinforcement learning, and take initial steps toward scalable methods that produce highly capable generalist agents. In contrast to vision and language domains, reinforcement learning has seen advocacy for the use of smaller models [16, 49, 8] and is usually either used to solve single tasks, or multiple tasks within the same environment. Importantly, training across multiple environments – with very different dynamics, rewards, visuals, and agent embodiments – has been studied less significantly.

---

[*]Equal contribution. `[leekh, ofirnachum, imordatch]@google.com`
[1]Additional information, videos and code can be seen at sites.google.com/view/multi-game-transformers.

36th Conference on Neural Information Processing Systems (NeurIPS 2022).

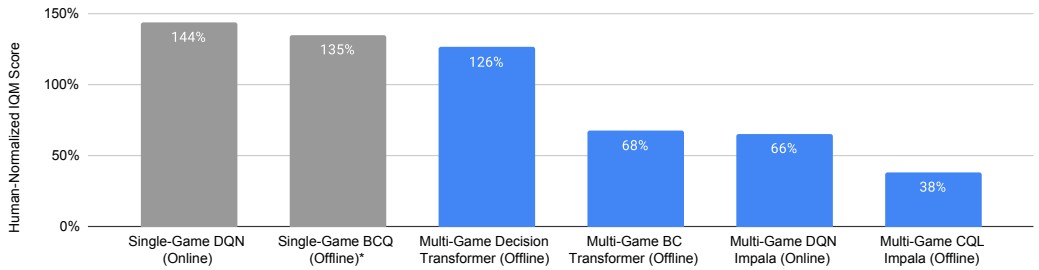

Figure 1: Aggregates of human-normalized scores (Inter-Quartile Mean) across 41 Atari games. Grey bars are single-game specialist models while blue are generalists. Single-game BCQ [21] results are from Gulcehre et al. [25]. Multi-game models are all trained on a dataset [1] with inter-quartile mean human-normalized score of 101%, which Multi-Game DT notably exceeds.

Specifically, we investigate whether a single model – with a single set of parameters – can be trained to act in multiple environments from large amounts of expert and non-expert experience. We consider training on a suite of 41 Atari games [9, 25] for their diversity, informally asking "Can models learn something universal from playing many video games?". To train this model, we use only the previously-collected trajectories from Agarwal et al. [1], but we evaluate our agent interactively. We are not striving for mastery or efficiency that game-specific agents can offer, as we believe we are still in early stages of this research agenda. Rather, we investigate whether the same trends observed in language and vision hold for large-scale generalist reinforcement learning agents.

We find that we can train a single agent that achieves 126% of human-level performance simultaneously across all games after training on offline expert and non-expert datasets (see Figure 1). Furthermore, we see similar trends that mirror those observed in language and vision: rapid fine-tuning to never-before-seen games with very little data (Section 4.5), a scaling relationship between performance and model size (Section 4.4), and faster training progress for larger models (Appendix G).

Notably, not all existing approaches to multi-environment training work well. We investigate several approaches, including treating the problem as offline decision transformer-based sequence modeling [14, 35], online RL [53], offline temporal difference methods [42], contrastive representations [56], and behavior cloning [60]. We find that decision transformer based models offer the best performance and scaling properties in the multi-environment regime. However, to permit training on both expert and non-expert trajectories, we find it is necessary to use a guided generation technique from language modeling to generate expert-level actions, which is an important departure from standard decision transformers.

Our contributions are threefold: First, we show that it is possible to train a single high-performing generalist agent to act across multiple environments from offline data alone. Second, we show that scaling trends observed in language and vision hold. And third, we compare multiple approaches for achieving this goal, finding that decision transformers combined with guided generation perform the best. It is our hope this study can inspire further research in generalist agents. To aid this, we make our pre-trained models and code publicly available.

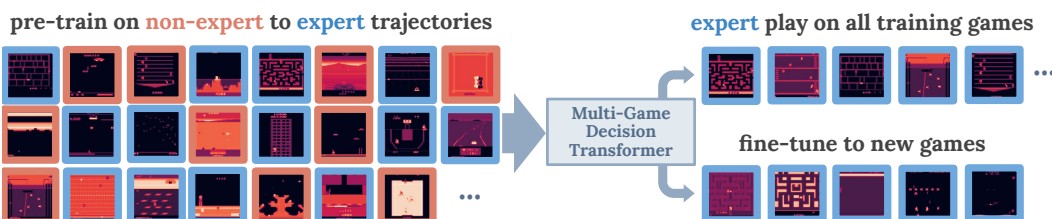

Figure 2: An overview of the training and evaluation setup. We observe expert-level game-play in the interactive setting after offline learning from trajectories ranging from beginner to expert.

## 2 Related Work

A generalist agent for solving a variety of environments has been a goal for artificial intelligence (AI) researchers since the inception of AI as a field of study [50]. This same reason motivated the introduction of the Atari suite (the Arcade Learning Environment, or ALE) as a testbed for learning algorithms [10]; in their own words, the ALE is for "empirically assessing agents designed for general competency." While the celebrated deep $Q$-learning [52] and actor critic [54] agents were among the first to use a single algorithm for all games, they nevertheless required separate training and hyperparameters for each game agent. Later works have demonstrated the ability to learn a single neural network agent on multiple Atari games simultaneously, either online [20] or via policy distillation [59, 67]. The aim of our work is similar – to learn a single agent for playing multiple Atari games – with a focus on offline learning. We demonstrate results with human-level competency on up to 46 games, which is unseen in the literature.

A closely related setting is learning to solve multiple tasks within the same or similar environments. For example in the robotics field, existing works propose to use language-conditioned tasks [48, 3, 34], while others posit goal-reaching as a way to learn general skills [51], among other proposals [37, 82]. In this work, we tackle the problem of learning to act in a large collection of environments with distinctively different dynamics, rewards, and agent embodiments. This complicated but important setting requires a different type of generalization that has been studied significantly less.

A concurrent work [65] also aims to train a transformer-based generalist agent based on offline data including for the ALE. This work differs from ours in that the offline training data is exclusively near-optimal and it requires prompting by expert trajectories at inference time. In contrast, we extend decision transformers [14] from the Upside-Down RL family [71, 68] to learn from a diverse dataset (expert and non-expert data), predict returns, and pick optimality-conditioned returns. Furthermore, we provide comparisons against existing behavioral cloning, online and offline RL methods, and contrastive representations [80, 56]. Other works that also consider LLM-like sequence modeling for a variety of single control tasks include [66, 84, 35, 23, 57].

## 3 Method

We consider a decision-making agent that at every time $t$ receives an observation of the world $\mathbf{o}^t$, chooses an action $a^t$, and receives a scalar reward $r^t$. Our goal is to learn a single optimal policy distribution $P_\theta^*(a^t|\mathbf{o}^{\leq t}, a^{<t}, r^{<t})$ with parameters $\theta$ that maximizes the agent's total future return $R^t = \sum_{k>t} r^k$ on all the environments we consider.

### 3.1 Reinforcement Learning as Sequence Modeling

Following [14], we pose the problem of offline reinforcement learning as a sequence modeling problem where we model the probability of the next sequence token $x_i$ conditioned on all tokens prior to it: $P_\theta(x_i|x_{<i})$, similar to contemporary decoder-only sequence models [12, 15, 62]. The sequences we consider have the form:

$$x = \langle ..., \mathbf{o}_1^t, ..., \mathbf{o}_M^t, \hat{R}^t, a^t, r^t, ... \rangle$$

where $t$ represents a time-step, $M$ is the number of image patches per observation (which we further discuss in Section 3.2), and $\hat{R}^t$ is the agent's target return for the rest of the sequence. Such a sequence order respects the causal structure of the environment decision process. Figure 3 presents an overview of our model architecture.

Returns, actions, and rewards are tokenized (See Section 3.2 for details), and we train the model to predict the next return, action, and reward discrete token in a sequence via standard cross-entropy loss. The sequence we consider is different from Chen et al. [14], which has $\langle ..., \hat{R}^t, \mathbf{o}^t, a^t, ... \rangle$. Our design allows predicting the return distribution and sampling from it, instead of relying on a user to manually select an expert-level return at inference time (See Section 3.4).

Predicting future value and rewards have been shown to be useful objectives for learning better representations in artificial reinforcement learning agents [47, 69, 44] and important signals for representation learning in humans [5]. Thus, while we may not directly use all of the predicted quantities, the task of predicting them encourages structure and representation learning of our

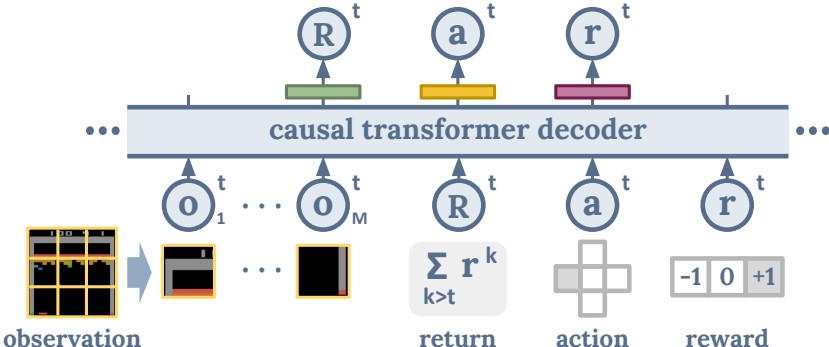

Figure 3: An overview of our decision transformer architecture.

environments. In this work, we do not attempt to predict future observations due to their non-discrete nature and the additional model capacity that would be required to generate images. However, building image-based forward prediction models of the environment has been shown to be a useful representation objective for RL [28, 27, 29]. We leave it for future investigation.

### 3.2 Tokenization

To generate returns, actions, and rewards via multinomial distributions similarly to language generation, we convert these quantities to discrete tokens. Actions $a$ are already discrete quantities in the environments we consider. We convert scalar rewards to ternary quantities $\{-1, 0, +1\}$, and uniformly quantize returns into a discrete range shared by all our environments.[2]

Inspired by the simplicity and effectiveness of transformer architectures for processing images [19], we divide each observation image into a collection of $M$ patches[3] (see Figure 3). Each patch is additively combined with a trainable position encoding and linearly projected into the input token embedding space. We experimented with using image tokenizations coming from a convolutional network, but did not find it to have a significant benefit and omitted it for simplicity.

We chose our tokenization scheme with simplicity in mind, but many other schemes are possible. While all our environments use a shared action space, varying action spaces when controlling different agent morphologies can still be tokenized using methods of [33, 43, 26]. And while we used uniform quantization to discretize continuous quantities, more sophisticated methods such as VQ-VAE [76] can be used to learn more effective discretizations.

### 3.3 Training Dataset

To train the model, we use an existing dataset of Atari trajectories (with quantized returns) introduced in [1]. The dataset contains trajectories collected from the training progress of a DQN agent [53]. Following [25], we select 46 games where DQN significantly outperforms a random agent. 41 games are used for training and 5 games are held out for out-of-distribution generalization experiments.

We chose 5 held-out games representing different categories including `Alien` and `MsPacman` (maze based), `Pong` (ball tracking), `SpaceInvaders` (shoot vertically), and `StarGunner` (shoot horizontally), to ensure out-of-distribution generalization can be evaluated on different types of games.

For each of 41 games, we use data from 2 training runs, each containing roll-outs from 50 policy checkpoints, in turn each containing 1 million environment steps. This totals 4.1 billion steps. Using the tokenization scheme in previous sections, the dataset contains almost 160 billion tokens.

As the dataset contains agent behaviors at all stages of learning, it contains both expert and non-expert behaviors. We do not perform any special filtering, curation, or balancing of the dataset. The

---

[2]The training datasets we use (Section 3.3) contains scalar reward values clipped to $[-1, 1]$. For return quantization, we use range $\{-20, ..., 100\}$ with bin size 1 in all our experiments as we find it covers most of the returns we observe in the datasets.

[3]We use 6x6 patches, where each patch corresponds to 14x14 pixels, in all our experiments.

motivation to train on such data instead of expert-only behaviors is twofold: Firstly, sub-optimal behaviors are more diverse than optimal behaviors and may still be useful for learning representations of the environment and consequences of poor decisions. Secondly, it may be difficult to create a single binary criteria for optimality as it is typically a graded quantity. Thus, instead of assuming only task-relevant expert behaviors, we train our model on all available behaviors, yet generate expert behavior at inference time as described in the next section.

### 3.4   Expert Action Inference

As described above, our training datasets contain a mix of expert and non-expert behaviors, thus directly generating actions from the model imitating the data is unlikely to consistently produce expert behavior (as we confirm in Section 4.7). Instead, we want to control action generation to consistently produce actions of highly-rewarding behavior. This mirrors the problem of discriminator-guided generation in language models, for which a variety of methods have been proposed [40, 79, 58].

We propose an inference-time method inspired by [40] and assume a binary classifier $P(\text{expert}^t|...)$ that identifies whether or not the behavior is expert-level before taking an action at time $t$. Following Bayes' rule, the distribution of expert-level returns at time $t$ is then:

$$P(\text{expert}^t|R^t, ...) \propto \exp(\kappa(R^t - R_{low})/(R_{high} - R_{low}))$$

where $R_{low}$ is the return lower bound and $R_{high}$ is the return upper bound. Similarly to [70, 73, 74, 39], we define a binary classifier to be proportional to future return with inverse temperature $\kappa$[4]:

$$P(\text{expert}^t|R^t, ...) \equiv \exp(\kappa R^t)$$

This results in a simple auto-regressive procedure where we first sample high-but-plausible target returns $R^t$ according to log-probability $\log P_\theta(R^t|...) + \kappa(R^t - R_{low})/(R_{high} - R_{low})$, and then sample actions according to $P_\theta(a^t|R^t, ...)$. See Figure 4 for an illustration of this procedure and Appendix B.3 for implementation details. It can be seen as a variation of return-conditioned policies [41, 71, 14] that automatically generates expert-level (but likely) returns at every timestep, instead of manually fixing them for the duration of the episode.

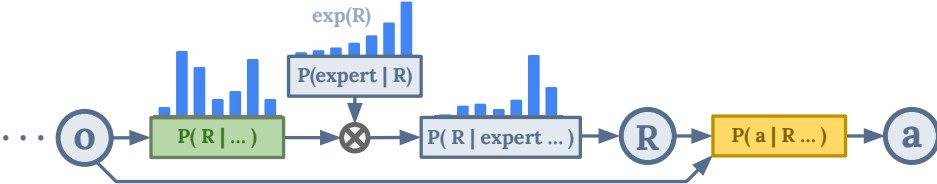

Figure 4: An illustration of our expert-level return and action sampling procedure. $P_\theta(R|...)$ and $P_\theta(a|R...)$ are the distributions learned by the sequence model.

Importantly, this formulation only affects the inference procedure of the model – training is entirely unaffected and can rely on standard next-token prediction frameworks and infrastructure. While we chose this formulation for its simplicity, controllable generation is an active area of study and we expect other more effective methods to be introduced in the future. As such, our contribution is to point out a connection between problems of controllable generation in language modeling and optimality conditioning in control.

## 4   Experiments

We formulate our experiments to answer a number of questions that are addressed in following sections:

- How do different **online and offline** methods perform in the multi-game regime?
- How do different methods **scale with model size**?
- How effective are different methods at **transfer to novel games**?

---

[4]We use $\kappa = 10$ in all our experiments.

- Does **multi-game decision transformer** improve upon training data?
- Does **expert action inference** (Section 3.4) improve upon behavioral cloning?
- Does training on **expert and non-expert data** bring benefits over expert-only training?

We also consider whether there are benefits to specifically using the **transformer architecture** in Appendix D, and qualitatively explore the attention behavior of these models in Appendix H.

## 4.1 Setup

**Model Variants and Scaling.** We base our decision transformer (DT) configuration on GPT-2 [12] as summarized in Appendix B.1. We report results for DT-200M (a Multi-Game DT with 200M parameters) if not specified otherwise. Other smaller variants are DT-40M and DT-10M. We set sequence length to 4 game frames for all experiments, which results in sequences of 156 tokens.

**Training and Fine-tuning.** We train all Multi-Game DT models on TPUv4 hardware and the Jaxline (Babuschkin et al. [7]) framework for 10M steps using the LAMB optimizer [81] with a $3 \cdot 10^{-4}$ learning rate, 4000 steps linear warm-up, no weight decay, gradient clip 1.0, $\beta_1 = 0.9$ and $\beta_2 = 0.999$, and batch size 2048. For fine-tuning on novel games, we train for 100k steps with a $10^{-4}$ learning rate, $10^{-2}$ weight decay and batch size of 256 instead. Both regimes used image augmentations as described in Appendix B.5.

**Metrics.** We measure performance on individual Atari games by human normalized scores (HNS) [53], i.e. $(\text{score} - \text{score}_{\text{random}}) / (\text{score}_{\text{human}} - \text{score}_{\text{random}})$, or DQN-normalized scores, i.e. normalizing by the best DQN scores seen in the training dataset instead of using human scores. To create an aggregate comparison metric across all games, we use inter-quartile mean (IQM) of human-normalized scores across all games, following evaluation best practices proposed in [2]. Due to the prohibitively long training times, we only evaluated one training seed. We additionally report median aggregate metric in Appendix E.

## 4.2 Baseline Methods

**BC** Our Decision Transformer (Section 3.1) can be reduced to a transformer-based Behavioral Cloning (BC) [60] agent by removing the target return condition and return token prediction. Similar to what we do for Decision Transformer, we also learn BC models at different scales (10M, 40M, 200M parameters) while keeping other configurations unchanged.

**C51 DQN** As a point of comparison for online performance, we use the C51 algorithm [11] which is a variant of deep $Q$-learning (DQN) but with a categorical loss for minimizing the temporal difference (TD) errors. Following improvements suggested in Hessel et al. [30] as well as our own empirical observations, we use multi-step learning with $n = 4$. For the single-game experiments, we use the standard convolutional neural network (CNN) used in the implementation of C51 [13]. For the multi-game experiments, we modify the C51 implementation based on a hyperparameter search to use an Impala neural network architecture [20] with three blocks using 64, 128, and 128 channels respectively with a batch size of 128 and update period of 256.

**CQL** For an offline TD-based learning algorithm we use conservative $Q$-learning (CQL) [42]. Namely, we augment the categorical loss of C51 with a behavioral cloning loss minimizing $-\log \pi_Q(a|s)$, where $(s, a)$ is a state-action pair sampled from the offline dataset and $\pi_Q(\cdot|s) = \text{softmax}(Q(s, \cdot))$. Following the recommendations in Kumar et al. [42] we weight the contribution of the BC loss by 1 when using 100% of the offline data (multi-game training) and 4 when using 1% (single-game finetuning). For scaling experiments, we vary the number of blocks and channels in each block of the Impala: the number of blocks and channels is one of (5 blocks, 128 channels) $\approx$ 5M params, (10 blocks, 256 channels) $\approx$ 30M params, (5 blocks, 512 channels) $\approx$ 60M params, (10 blocks, 512 channels) $\approx$ 120M params.

**CPC, BERT, and ACL** For rapid adaptation to new games via fine-tuning, we consider representation learning baselines including contrastive predictive coding (CPC) [56], BERT pretraining [18], and attentive contrastive learning (ACL) [80]. All state representation networks are implemented

as additional multi-layer perceptrons (MLPs) or transformer layers on top of the Impala CNN used in C51 and CQL baselines. CPC uses two additional MLP layers with $512$ units each interleaved with `ReLU` activation to represent $\phi(s)$, which is optimized by maximizing $\phi(s)^\top W \phi(s')$ of true transitions $(s, s')$ and minimizing $\phi(s)^\top W \phi(\tilde{s})$ where $\tilde{s}$ is a state randomly sampled from the batch (including states from other games). For BERT pretraining, we use 2 self-attention layers with 4 attention heads of 256 units each and feed-forward dimension $512$, and train $\phi(s)$ using BERT's masked self-prediction loss on a trajectory of sequence length 16. ACL shares the same model parametrization as BERT, with the inclusion of action prediction in the pretraining objective.

### 4.3 How do different online and offline methods perform in the multi-game regime?

We compare different online and offline algorithms in the multi-game regime and their single-game counterparts in Figure 1. We find that single-game specialists are still most performant. Among multi-game generalist models, our Multi-Game Decision Transformer model comes closest to specialist performance. Multi-game online RL with non-transformer models comes second, while we struggled to get good performance with offline non-transformer models. We note that our multi-game online C51 DQN median score of 68% (see Appendix E) which compares similarly to multi-game median Impala score of 70%, which we calculated from results reported by [20] for our suite of games.

We believe the apparent advantage of offline DT compared to online multi-game methods like C51 may be explained in part through classical differences between online and offline settings in RL [45]. Online methods must balance exploration with the ability to learn and generalize from experience, which could be challenging in the multi-game setting, whereas offline DT only needs to learn to distill and generalize from the fixed multi-game experience given to it (collected by specialist DQN agents [1]). Beyond the difference between online and offline, one could also argue that C51 suffers from more training instability than DT due to the use of a temporal difference (TD) loss, which we discuss in the next paragraph.

### 4.4 How do different methods scale with model size?

In large language and vision models, lowest-achievable training loss typically decreases predictably with increasing model size. Kaplan et al. [38] demonstrated an empirical scaling relationship between the capacity of a language model (NLP terminology for a next-token autoregressive generative model) and its performance (negative log likelihood on held-out data). These trends were verified over many orders of magnitude of model size, ranging from few-million parameter models to hundreds of billion parameter models.

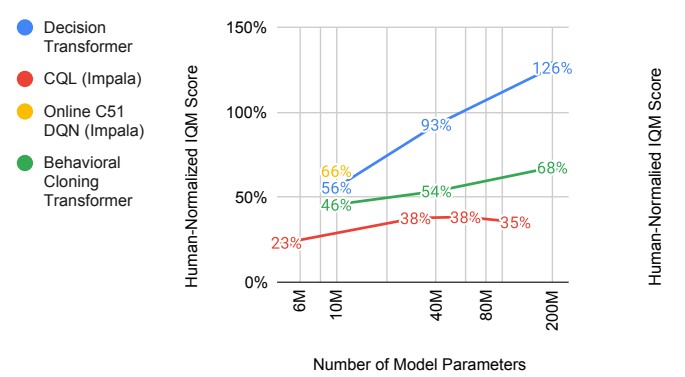
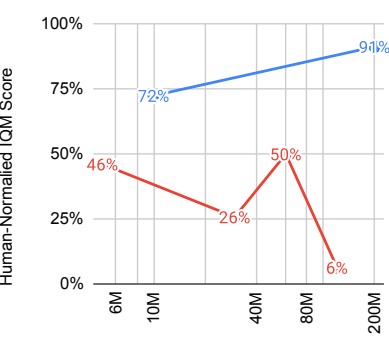

(a) Scaling of IQM scores for all training games with different model sizes and architectures.

(b) Scaling of IQM scores for all novel games after fine-tuning DT and CQL.

Figure 5: How model performance scales with model size, on training set games and novel games. (Impala) indicates using the Impala CNN architecture.

We investigate whether similar trends hold for *interactive* in-game performance – not just training loss – and show a similar performance scaling trend in Figure 5a. Multi-Game Decision Transformer

performance reliably increases over more than an order of magnitude of parameter scaling, whereas the other methods either saturate, or have much slower performance growth.

In contrast, in Figures 5a and 5b, we find that CQL does not improve with increased model size, and actually shows a sharp drop in the performance of larger models on the fine-tuning tasks. Temporal Difference (TD) methods suffer greater instability with larger model size in the multi-game setting, leading to this "inverse" scaling. Indeed, our attempts at other objectives closer to pure TD (C51, DQN, DDQN) led to even worse results (which we do not report). We note that similar conclusions about instability with respect to network size have been made by other work [22].

We also find that larger models train faster, in the sense of reaching higher in-game performance after observing the same number of tokens. We discuss these results in Appendix G.

## 4.5 How effective are different methods at transfer to novel games?

Pretraining for rapid adaptation to new games has not been explored widely on Atari games despite being a natural and well-motivated task due to its relevance to how humans transfer knowledge to new games. Nachum and Yang [55] employed pretraining on large offline data and fine-tunining on small expert data for Atari and compared to a set of state representation learning objectives based on bisimulation [24, 83], but their pretraining and fine-tuning use the same game. We are instead interested in the *transfer* ability of pretrained agents to new games.

We hence devise our own evaluation setup by pretraining DT, CQL, CPC, BERT, and ACL on the full datasets of the 41 training games with 100M steps each, and fine-tuning one model per held-out game using 1% (1M steps) from each game. The 1% fine-tuning data is uniformly sampled from the 50M step dataset without quality filtering. DT and CQL use the same objective for pretraining and fine-tuning, whereas CPC, BERT, and ACL each use their own pretraining objective and are fine-tuned using the BC objective. All methods are fine-tuned for 100,000 steps, which is much shorter than training any agent from scratch. We additionally include training CQL from scratch on the 1% held-out data to highlight the benefit of rapid fine-tuning.

Fine-tuning performance on the held-out games is shown in Figure 6. Pretraining with the DT objective performs the best across all games. All methods with pretraining outperform training CQL from scratch, which verifies our hypothesis that pretraining on other games should indeed help with rapid learning of a new game. CPC and BERT underperform DT, suggesting that learning state representations alone is not sufficient for desirable transfer performance. While ACL adds an action prediction auxiliary loss to BERT, it showed little effect, suggesting that modeling the actions in the right way on the offline data is important for good transfer performance. Furthermore, we find that fine-tuning performance improves as the DT model becomes larger, while CQL fine-tuning performance is inconsistent with model size (see Figure 5b).

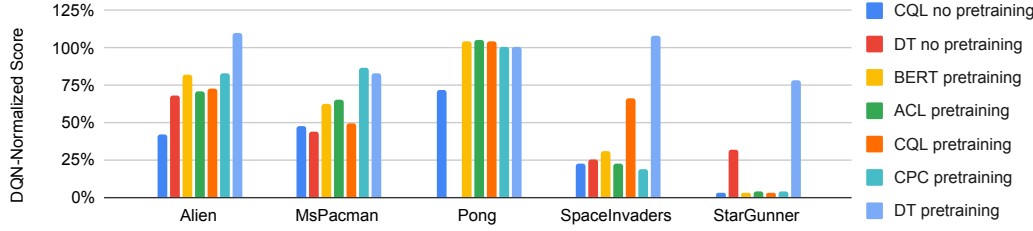

Figure 6: Fine-tuning performance on 1% of 5 held-out games' data after pretraining on other 41 games using DT, CQL, CPC, BERT, and ACL. All pretraining methods outperform training CQL from scratch on the 1% held-out data, highlighting the transfer benefit of pretraining on other games. DT performs the best among all methods considered.

## 4.6 Does multi-game decision transformer improve upon training data?

We want to evaluate whether decision transformer with expert action inference is capable of acting better than the best demonstrations seen during training. To do this, we look at the top 3 performing decision transformer model rollouts. We use top 3 rollouts instead of the mean across all rollouts

to more fairly compare to the *best* demonstration, rather than an average expert demonstration. We show percentage improvement over best demonstration score for individual games in Figure 7. We see significant improvement over the training data in a number of games.

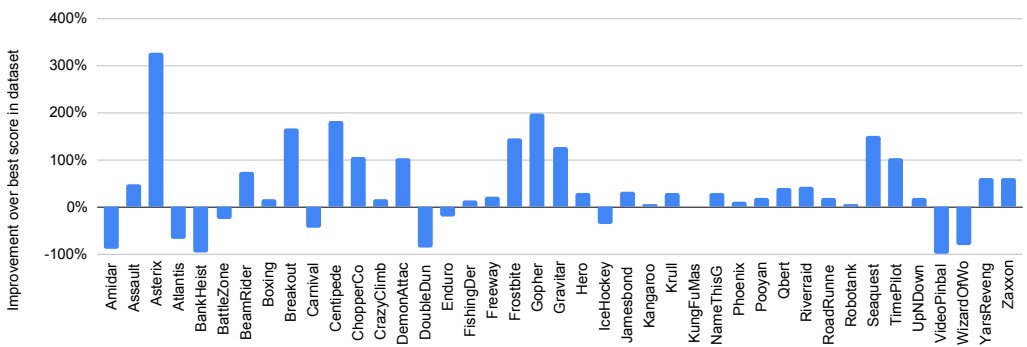

Figure 7: Percent of improvement of top 3 decision transformer rollouts over the best score in the training dataset. 0% indicates no improvement. Top-3 metric (instead of mean) is used to more fairly compare to the best – rather than expert average – demonstration score.

## 4.7 Does optimal action inference improve upon behavior cloning?

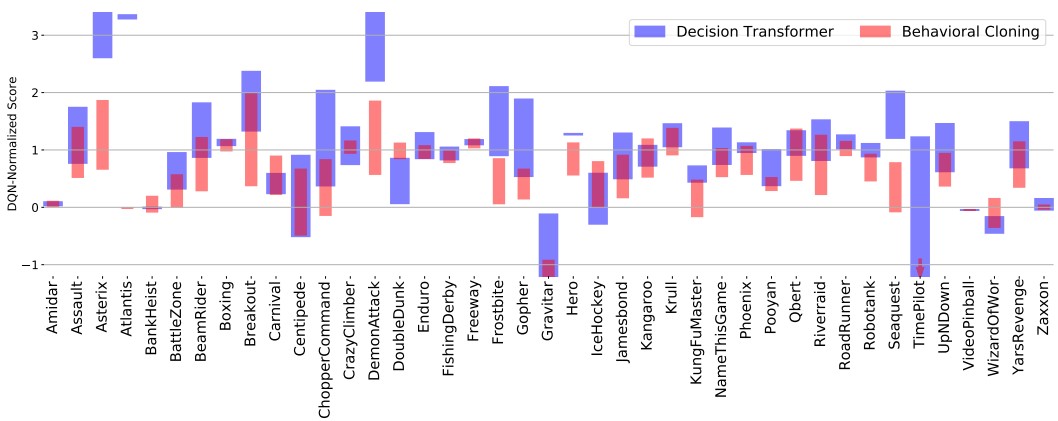

Figure 8: Comparison of per-game scores for decision transformer to behavioral cloning. Bars indicate ± standard deviation around the mean across 16 trials. We show DQN-normalized scores in this figure for better presentations.

In Figure 1 we see that IQM performance across all games is indeed significantly improved by generating optimality-conditioned actions. Figure 8 shows the mean and standard deviation of scores across all games. While behavior cloning may sometimes produce highly-rewarding episodes, it is less likely to do so. We find decision transformer outperforms behavioral cloning in 31 out of 41 games.

## 4.8 Does training on expert and non-expert data bring benefits over expert-only training?

We believe that, comparing to learning from expert demonstrations, learning from large, diverse datasets that include some expert data but primarily non-expert data help learning and improve performance. To verify this hypothesis, we filter our training data [1] from each game by episodic returns and only preserve top 10% trajectories to produce an expert dataset (see Appendix F for details). We use this expert dataset to train our multi-game decision transformer (DT-40M) and the transformer-based behavioral cloning model (BC-40M). Figure 9 compares these models trained on expert data and our DT-40M trained on all data.

We observe that (1) Training only on expert data improves behavioral cloning; (2) Training on full data, including expert and non-expert data, improves Decision Transformer; (3) Decision Transformer with full data outperforms behavioral cloning trained on expert data.

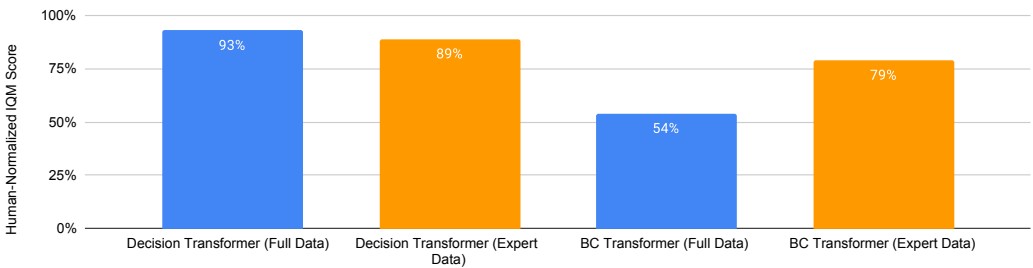

Figure 9: Comparison of 40M transformer models trained on full data and only expert data.

## 5    Conclusion

In the quest to develop highly capable and generalist agents, we have made important and measurable progress. Namely, our results exhibit a clear benefit of using large transformer-based models in multi-game domains, and the general trends in these results – performance improvements with larger models and the ability to rapidly fine-tune to new tasks – mirror the successes observed for large-scale vision and language models. Our results also highlight difficulties of online RL algorithms in handling the complexity of multi-game training on Atari. It is interesting to note that our best results are achieved by decision transformers, which essentially learn via supervised learning on sequence data, compared to alternative approaches such as temporal difference learning (more typical in reinforcement learning), policy gradients, and contrastive representation learning. This begs the question of whether online learning algorithms can be modified to be as "data-absorbent" as DT-like methods. While even our best generalist agents at times fall short of performance achieved by agents trained on a single task, this is broadly consistent with related works that have trained single models on many tasks [36, 65]. Still, our best generalist agents are already capable of outperforming the data they are trained on. We believe the trends suggest clear paths for future work – that, with larger models and larger suites of tasks, performance is likely to scale up commensurately.

**Limitations.** We acknowledge reasons for caution in over-generalizing our conclusions. Our results are based largely on performance in the Atari suite, where action and observation spaces are aligned across different games. It is unclear whether offline RL datasets such as Atari are of sufficient scale and diversity that we would see similar performance scaling as observed in NLP and vision benchmarks. Whether we can observe other forms of generalization, such as zero-shot adaptation, as well as whether our conclusions hold for other settings, remains unclear.

**Societal Impacts.** In the current setting, we do not foresee significant societal impact as the models are limited to playing simple video games. We emphasize that our current agents are not intended to interact with humans or be used outside of self-contained game-playing domains. One should exercise increased caution if extending our algorithms and methods to such situations in order to ensure any safety and ethical concerns are appropriately addressed. At the same time, the capability of decision making based on reward feedback – rather than purely imitation of the data – has the potential to be easier to align with human values and goals.

## Acknowledgements

We would like to thank Oscar Ramirez, Roopali Vij, Sabela Ramos, Rishabh Agarwal, Shixiang (Shane) Gu, Aleksandra Faust, Noah Fiedel, Chelsea Finn, Sergey Levine, John Canny, Kimin Lee, Hao Liu, Ed Chi, and Luke Metz for their valuable contributions and support for this work.

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
