# A Contribution Statement

**Kuang-Huei Lee** Proposed project direction. Contributed to the JAX and TF Decision Transformer (DT) and BC code. Ran BC and DT experiments. Contributed to paper writing.

**Ofir Nachum** Proposed project direction. Contributed to TF code for DQN, CQL, and representation learning algorithms. Ran experiments for DQN and CQL. Contributed to paper writing.

**Mengjiao Yang** Contributed code in representation learning algorithms. Ran experiments for CPC, BERT, ACL pretraining + OOD finetuning. Contributed to paper writing.

**Lisa Lee** Contributed to TF codebase. Ran alternative environment experiments. Helped with paper editing.

**Daniel Freeman** Contributed to JAX training code and dataset generation pipelines. Generated datasets. Helped with paper writing.

**Winnie Xu** Contributed to JAX training infrastructure, beginner / expert dataset generation, augmentation and metrics pipelines. Ran experiments for alternative DT variants. Helped with paper writing. Work done while a Research Intern.

**Sergio Guadarrama** Helped with project direction and experiment discussions. Worked on paper editing.

**Ian Fischer** Helped with project direction and experiment discussions. Worked on paper editing.

**Eric Jang** Helped with project direction, contributed to building JAX infrastructure. Helped with paper writing.

**Henryk Michalewski** Contributed to building jaxline training and data processing infrastructure. Ran fine-tuning experiments. Helped with project direction and experiment discussions. Contributed to paper writing.

**Igor Mordatch** Proposed project direction. Contributed to building JAX training and data processing infrastructure. Ran DT experiments. Contributed to paper writing and visualizations.

# B Implementation Details

## B.1 Transformer network architecture

The input consists of a sequence of observations, returns, actions and rewards. Observations are images in the format $B \times T \times W \times H \times C$. We use $84 \times 84$ grayscale images (*i.e.*, $W = 84, H = 84, C = 1$). Similar to ViT [19], we extract $M$ non-overlapping image patches, perform a linear projection and then rasterise them into $d_{model}$-dimensional 1D tokens. We define each patch to be $14 \times 14$ pixels (*i.e.*, $M = 6 \times 6 = 36$). A learned positional embedding is added to each of the patch tokens $\mathbf{o}_1, ..., \mathbf{o}_M$ to retain positional information as in ViT. As described in Section 3.2, returns are discretized into 120 buckets in $\{-20, ..., 100\}$, and rewards are converted to ternary quantities $\{-1, 0, +1\}$.

For the whole sequence $\langle ..., \mathbf{o}_1^t, ..., \mathbf{o}_M^t, \hat{R}^t, a^t, r^t, ... \rangle$, we learn another positional embedding at each position and add to each token embedding. We experimented with rotary position embedding [72], but did not find a significant benefit from them in our setting. On top of the token embeddings, our transformer models use a standard transformer decoder architecture.

A standard transformer implementation for sequence modeling would employ a sequential causal attention masking to prevent positions from attending to subsequent positions [77]. However, for the sequence $\langle ..., \mathbf{o}_1^t, ..., \mathbf{o}_M^t, \hat{R}^t, a^t, r^t, ... \rangle$ that we consider, we do not want to prevent the position corresponding to observation token $\mathbf{o}_m^t$ from accessing subsequent observation tokens $\{\mathbf{o}_{m'}^t : m' > m\}$ within the same timestep, since there is no clear sequential causal relation between image patches. Therefore, we change the sequential causal masking to allow observation tokens within the same timestep to access each other, but not subsequent positions after $\mathbf{o}_M^t$, *i.e.* $\hat{R}^t, a^t, r^t, \mathbf{o}_1^{t+1}, ..., \mathbf{o}_M^{t+1}, ...$

Table 1 summarizes the transformer configurations we use for each model size. We train these models on an internal cluster, each with 64 TPUv4. Due to prohibitively long training times, we only evaluated one training seed. For the 40M DT model, we perform four additional runs with different

seeds to investigate the sensitivity to random seeds. The mean and standard deviation of the median score across 5 random seeds are 0.79 and 0.06; the mean and standard deviation of IQM scores are 0.95 and 0.05. Compared to the differences to the baseline results and the results of other DT model sizes, we consider the variance to be relatively small.

| Model | Layers | Hidden size $D$ ($d_{model}$) | Heads | Params | Training Time on 64 TPUv4 |
|---|---|---|---|---|---|
| DT-10M | 4 | 512 | 8 | 10M | 1 day |
| DT-40M | 6 | 768 | 12 | 40M | 2 days |
| DT-200M | 10 | 1280 | 20 | 200M | 8 days |

Table 1: Multi-Game Decision Transformer Variants

We performed hyperparameter search on DT-40M, and directly applied to DT-200M. With the set of hyperparameters working the best for smaller models, it still manifests a clear scaling trend of multi-game and transfer learning performance.

## B.2  Fine-tuning protocol for Atari games

In the fine-tuning experiments, we reserved five games (Alien, MsPacman, Pong, Space Invaders and Star Gunner) to be used only for fine-tuning. These games were selected due to their varied gameplay characteristics. Each game was fine-tuned separately to measure the model's transfer performance for a fixed game. We use 1% of the original dataset (corresponding to roughly $500\,000$ transitions) to specifically test fine-tuning in low-data regimes.

## B.3  Action and return sampling during in-game evaluation

We sample actions from the model with a temperature of 1. Inspired by Nucleus sampling (Holtzman et al. [32]), we only sample from the top 85th percentile action logits for all Decision Transformer models and Behavioral Cloning models (this parameter was selected to give highest performance for both models). While we train the model to predict actions for all timesteps in the sequence, during in-game evaluation, we execute the last predicted action in the sequence (conditioned on all past observations, and past generated actions, rewards, and target returns).

To generate target returns as discussed in Section 3.4, we sample them from the model with the temperature of 1 and the top 85th percentile logits. We use $\kappa = 10$ in all our experiments. To avoid storing the history of previously generated target returns (which may be difficult to incorporate into some RL frameworks), we experimented with autoregressively regenerating all target returns in the sequence, and found that to work well without requiring any special recurrent state maintenance outside of the model. Algorithm 1 has pseudocode for our expert return and action inference.

As an alternative way to generate expert-but-likely returns, we also experimented with simply generating $N$ return samples from the model according to log-probability $\log P_\theta(R^t|...)$, and picking the highest one. We then generate the action conditioned on this largest picked return as before. This avoids needing the hyperparameter $\kappa$. In this setting, we found $N = 128$, inverse temperature of 0.75 for return sampling, no percentile cutoff for return sampling, and sampling from the top 50th percentile action logits with a temperature of 1 to work similarly well.

## B.4  Evaluation protocol and Atari environment details

Our environment is the Atari 2600 Gym environment with pre-processing performed as in Agarwal et al. [1]. Our Atari observations are $84 \times 84$ grayscale images. We compress observation images to jpeg in the dataset (to keep dataset size small) and during in-game evaluation. All games use the same shared set of 18 discrete actions. For all methods, each game score is calculated by averaging over 16 model rollout episode trials. To reduce inter-trial variability, we do not use sticky actions during evaluation.

## B.5  Image augmentation

All models were trained with image augmentations. We investigate training with the following augmentation methods: random cropping, random channel permutation, random pixel permutation,

---

**Algorithm 1** Pseudocode for Expert Return and Action Inference (Section 3.4)

---

Given an environment $E$, the current time step $t$. $\kappa = 10$, the return upper bound $R_{high} = 100$, the return lower bound $R_{low} = -20$

Initialize a context window $\langle ..., r^{t-2}, \mathbf{o}_1^{t-1}, ..., \mathbf{o}_M^{t-1}, \hat{R}^{t-1}, a^{t-1}, r^{t-1}, \mathbf{o}_1^t, ..., \mathbf{o}_M^t \rangle$, abbreviated as ... in the following.

*# Autoregressive return and action generation*

**while** terminal state not reached yet **do**

    Compute 121 logits ($R^t = -20, \ldots, 100$) for the categorical return distribution $P(R^t|...)$

    *# Increase logits proportionally to return magnitudes to prefer high magnitude*

    Define $\log P(R^t|\text{expert}^t, ...) = \log P(R^t|...) + \kappa(R^t - R_{low})/(R_{high} - R_{low})$

    *# Sample a return*

    $R^t \sim P(R^t|\text{expert}^t, ...)$

    Compute logits for the categorical action distribution $P(a^t|R^t, ...)$

    *# Sample an action*

    $a^t \sim P(a^t|R^t, ...)$

    *# Interact with the environment*

    $\mathbf{o}_1^{t+1}, ..., \mathbf{o}_M^{t+1}, r^t \sim E_{step}(a^t)$

    $t = t + 1$ and shift the context window accordingly

**end while**

---

horizontal flip, vertical flip, and random rotations. We found random cropping and random rotations to work the best. (In our random cropping implementation, images of size $84 \times 84$ are padded on each side with 4 zero-value pixels, and then randomly cropped to $84 \times 84$.) In general, we aim to expand the domain of problems solved during training to similar kinds that we hope to generalize to by encoding useful inductive biases. We maintain the same random augmentation parameters for each window sequence. We apply data augmentation in both pre-training and fine-tuning.

## C    Baseline Implementation Details

**BC**    Our BC model is effectively the same as our DT model but removing the return token $\hat{R}^t$ from the training sequence:

$$x = \langle ..., \mathbf{o}_1^t, ..., \mathbf{o}_M^t, a^t, r^t, ... \rangle$$

Instead of predicting a return token (distribution) given observation tokens $\mathbf{o}_1^t, ..., \mathbf{o}_M^t$ and the previous part of the sequence, we directly predict an action token (distribution), which also means that we remove return conditioning for the BC model. During evaluation, we sample actions with a temperature of 1, and sample from the top 85th percentile logits (as discussed in Appendix B.3). All other implementation details and configurations are identical to DT.

**C51 DQN**    For single-game experiments, our implementation and training followed the details in [11] except for using multi-step learning with $n = 4$. For multi-game experiments we trained using the details provided in the main text; we ran the algorithm for 15M gradient steps ($\approx$ 4B environment steps $\approx$ 16B Atari frames).

**CQL**    For CQL we use the same optimizer and learning rate as for C51 DQN. We use a per-replica batch size of 32 and run for 1M gradient steps on a TPU pod with 32 cores, yielding a global batch size of 256. During finetuning for each game, we copy the entire $Q$-network trained with CQL, and apply an additional 100k gradient steps of batch size 32 on a single CPU, where each batch is sampled exclusively from the offline dataset of the finetuned game. We also experimented with smaller learning rates (0.00003 instead of the default 0.00025) and larger batch sizes (1024, 4096) but found the results largely unchanged. We also tried using offline C51 and double DQN as opposed to CQL, and found performance to be worse.

**CPC**    For the CPC baseline [56], we apply a contrastive loss between $\phi(o_t), \phi(o_{t+1})$ using the objective function

$$- \phi(o_{t+1})^\top W \phi(o_t) + \log \mathbb{E}_{\tilde{s} \sim \rho}[\exp\{\phi(\tilde{o})^\top W \phi(o_t)\}], \tag{1}$$

where $W$ is a trainable matrix and $\rho$ is a non-trainable prior distribution; for mini-batch training we set $\rho$ to be the distribution of states in the mini-batch. The state representations $\phi(o)$ is parametrized by CNNs followed by two MLP layers with 512 units each interleaved with ReLU activation. For the CNN architecture, we used the C51 implementation with an Impala neural network architecture of three blocks using 16, 32, and 32 channels respectively, and trained with a batch size of 256 and learning rate of 0.00025 both during pretraining and downstream BC adaptation. We conduct representation learning for a total of 1M gradient steps, and finetune on 1% data for 100k steps every 50k steps of representation learning and report the best finetuning results.

**BERT and ACL**  Our BERT and ACL baselines are based on the representation learning objectives described in [80]. For the BERT [18] state representation learning baseline, we (1) take a sub-trajectory $o_{t:t+k}, a_{t:t+k}, r_{t:t+k}$ from the dataset (without special tokenization as in DT), (2) randomly mask a subset of these, (3) pass the masked sequence into a transformer, and then (4) for each masked input state $o_{t+i}$, apply a contrastive loss between its representation $\phi(o_{t+i})$ and the transformer output $\text{Transformer}[i]$ at the corresponding sequence position:

$$- \phi(o_{t+i})^\top W \, \text{Transformer}[i] + \log \mathbb{E}_{\tilde{o} \sim \rho}[\exp\{\phi(\tilde{o})^\top W \, \text{Transformer}[i]\}], \qquad (2)$$

where $\rho$ is the distribution over states in the mini-batch. For attentive contrastive learning (ACL) [80], we apply an additional action prediction loss to the output of BERT at the sequence positions of the action inputs.

To parameterize $\phi$, we use the same CNN architecture as in CPC, while the transformer is parameterized by two self-attention layers with 4 attention heads of 256 units each and feed-forward dimension 512. The transformer does not apply any additional directional masking to its inputs. We used $k = 16$.

Pretraining and finetuning is analogous to CPC. Namely, when finetuning we take the pretrained representation $\phi$ and use a BC objective for learning a neural network (two MLP layers with 512 units each) policy on top of this representation.

## D    Comparisons between transformers and convolution networks

Decision Transformer is an Upside-Down RL (UDRL) [68, 71] implementation that uses the transformer architecture and considers RL as a sequence modeling problem. To understand the benefit of the transformer architecture, we compare to a UDRL implementation that uses feed-forward, convolutional Impala networks [20]. Specifically, we use the same return, action, and reward tokenizers as in DT, and only replace the observation (four consecutive Atari frames stacked together) encoding to use the Impala architecture. Similar to what we do for CQL, we also experiment with different sizes of the Impala architecture by varying the number of blocks and channels in each block of the Impala network: the number of blocks and channels is one of $(5 \text{ blocks}, 128 \text{ channels}) \approx 5\text{M params}$, $(10 \text{ blocks}, 256 \text{ channels}) \approx 30\text{M params}$, $(5 \text{ blocks}, 512 \text{ channels}) \approx 60\text{M params}$. We use a $(768, 768)$ 2-layer fully-connected head to predict the next return token from observation embedding; another $(768, 768)$ head to predict the next action token from a concatenation of observation embedding and return token embedding; and another

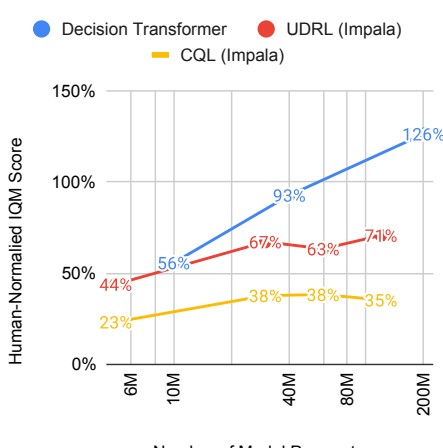

Figure 10: Performance scaling with model size for UDRL and CQL (Impala architecture) compared to Decision Transformer.

$(768, 768)$ head to predict the next reward token from a concatenation of observation embedding, return token embedding, and action token embedding.

The input to the model is slightly different from what we have for DT: Instead of considering a $T$-timestep sub-trajectory ($T = 4$) where each timestep contains $\mathbf{o}^t, R^t, a^t, r^t$, we stack $T$ image

frames (as common in [53]), and only consider $R^t, a^t, r^t$ from the last timestep. All other design choices and evaluation protocols are the same as DT.

Figure 10 shows clear advantages of Decision Transformer over UDRL with the Impala architecture. In the comparison between UDRL (Impala) and CQL that uses the same Impala network at each model size we evaluated, we observe that UDRL (Impala) outperforms CQL. The results show that the benefits of our method come not only from using network architectures, but also from the UDRL formulation. While the reasons for the benefits of transformer architecture over other neural networks are still an open question in general (see e.g., [64]), our hypothesis is that transformers allow for easier discovery of correlations between components of the input and output, due to the fact that transformers process the input as a flat sequence with attention allowed between any patch, action, or return token. Although it is not feasible to compare transformer with all possible convolutional architectures due to the broad design space, we believe these empirical results still show a clear trend favoring both UDRL and transformer architectures.

# E   Comparisons between methods using median human normalized scores

We used inter-quartile mean (IQM) to aggregate performance over individual games in Figure 1. Median is another metric commonly used to aggregate scores (although it has issues as discussed in [2]: it has high variability, and in the most extreme case, the median is unaffected by zero performance on nearly half of the tasks.). For completeness, we report the median scores for all methods:

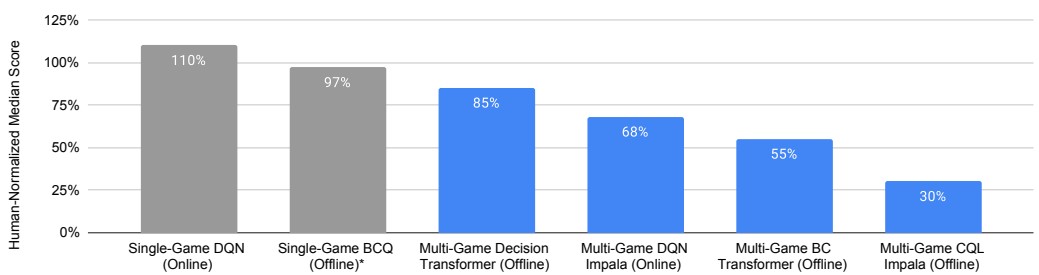

Figure 11: Median human-normalized score across 41 Atari games. Grey bars are single-game specialist models while blue are generalists. Single-game BCQ results are from Gulcehre et al. [25].

For expert-filtering experiments in Section 4.8, we also provide the plot of expert filtering effects with median human-normalized scores in Figure 12. We note that ranking of various configurations do not change across aggregate metrics.

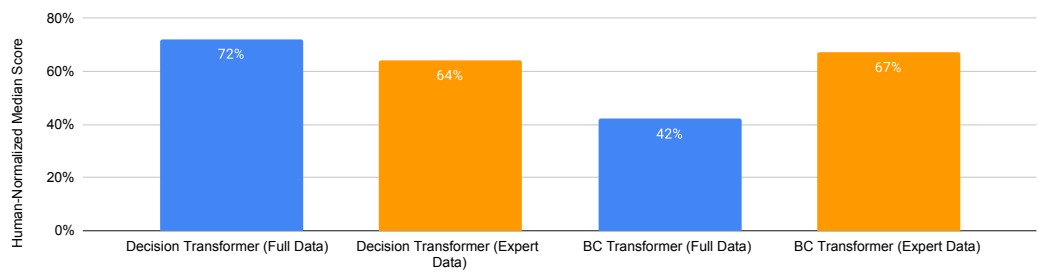

Figure 12: Median human-normalized scores of 40M transformer models trained on full data and only expert data.

For Upside-Down RL comparison experiments Appendix D, we also provide median human-normalized scores in Figure 13.

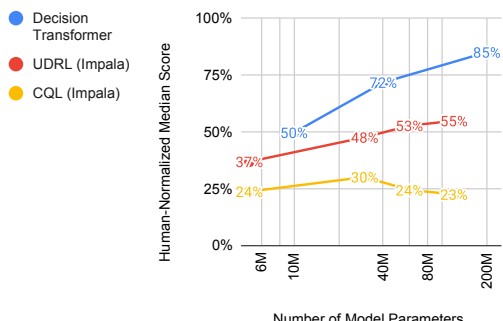

Figure 13: How UDRL (Impala architecture) median human-normalized score scales with model size on training set games, in comparisons with Decision Transformer and CQL (Impala architecture).

## F    Details of Expert Dataset Generation

To generate the expert dataset for experiments in Section 4.8, we we filter our training data [1] from each game by episodic returns and only preserve top 10% trajectories to produce an expert dataset. We plot of return histograms for reference in Figure 14.

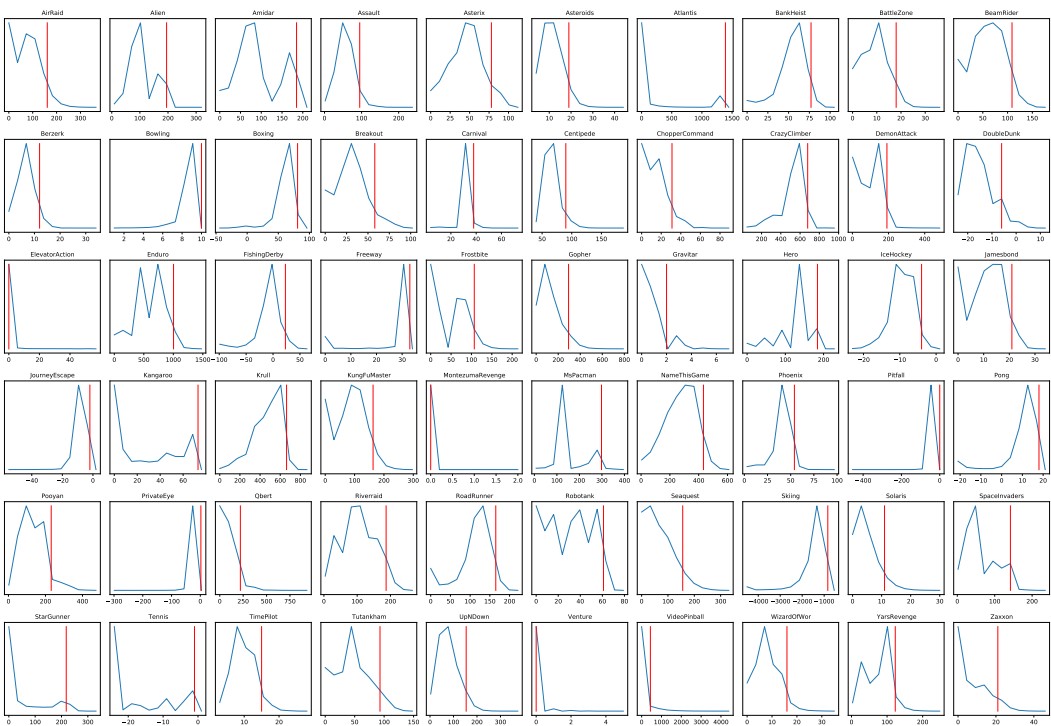

Figure 14: Histograms of rollout performance from [1] used to generate the expert dataset, with (unnormalized) score-density on the vertical axis, and game score (rewards are clipped) on the horizontal axis. We indicate the 90th percentile performance cutoff with a red vertical line for each game. Rollouts that exceeded this score threshold were included in the expert dataset.

## G    Effect of Model Size on Training Speed

It is believed that large transformer-based language models train faster than smaller models, in the sense that they reach higher performance after observing a similar number of tokens [38, 15]. We

find this trend to hold in our setting as well. Figure 15 shows an example of performance on two example games as multi-game training progresses. We see that larger models reach higher scores per number of training steps taken (thus tokens observed).

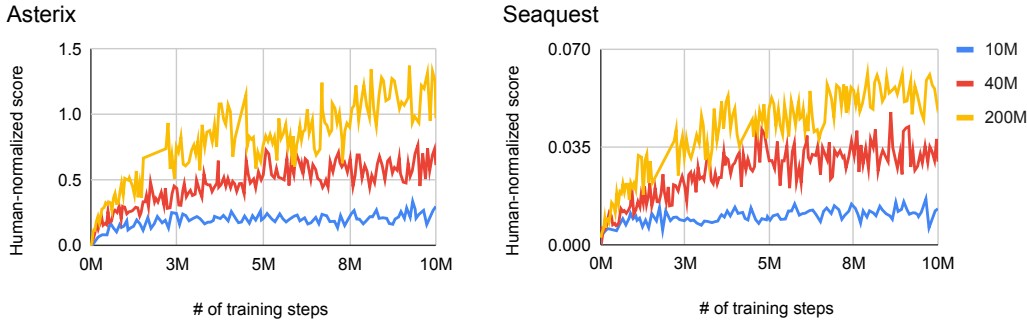

Figure 15: Example game scores for different model sizes as multi-game training progresses.

## H  Qualitative Attention Analysis

We find that the Decision Transformer model consistently attends to observation image patches that contain meaningful game entities. Figure 16 visualizes selected attention heads and layers for various games. We find heads consistently attend to entities such as player character, player's free movement space, non-player objects, and environment features.

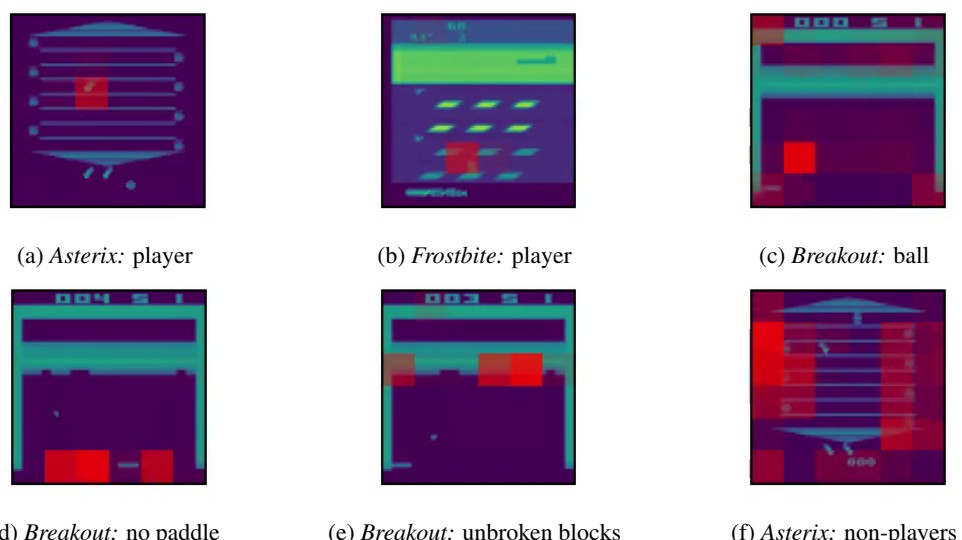

Figure 16: Example image patches attended (red) for predicting next action by Decision Transformer.

# I  Raw Atari Scores

We report full raw scores of 41 training Atari games for best performing sizes of multi-game models in Table 2.

| Game Name | DT (200M) | BC (200M) | Online DQN (10M) | CQL (60M) |
|---|---|---|---|---|
| Amidar | 101.5 | 101.0 | 629.8 | 4.0 |
| Assault | 2,385.9 | 1,872.1 | 1,338.7 | 820.1 |
| Asterix | 14,706.3 | 5,162.5 | 2,949.1 | 950.0 |
| Atlantis | 3,105,342.3 | 4,237.5 | 976,030.4 | 16,800.0 |
| BankHeist | 5.0 | 63.1 | 1,069.6 | 20.0 |
| BattleZone | 17,687.5 | 9,250.0 | 26,235.2 | 5,000.0 |
| BeamRider | 8,560.5 | 4,948.4 | 1,524.8 | 3,246.4 |
| Boxing | 95.1 | 90.9 | 68.3 | 100.0 |
| Breakout | 290.6 | 185.6 | 32.6 | 62.0 |
| Carnival | 2,213.8 | 2,986.9 | 2,021.2 | 440.0 |
| Centipede | 2,463.0 | 2,262.8 | 4,848.0 | 2,904.0 |
| ChopperCommand | 4,268.8 | 1,800.0 | 951.4 | 400.0 |
| CrazyClimber | 126,018.8 | 123,350.0 | 146,362.5 | 139,300.0 |
| DemonAttack | 23,768.4 | 7,870.6 | 446.8 | 1,202.0 |
| DoubleDunk | -10.6 | -1.5 | -156.2 | -2.0 |
| Enduro | 1,092.6 | 793.2 | 896.3 | 729.0 |
| FishingDerby | 11.8 | 5.6 | -152.3 | 18.4 |
| Freeway | 30.4 | 29.8 | 30.6 | 32.0 |
| Frostbite | 2,435.6 | 782.5 | 2,748.4 | 408.0 |
| Gopher | 9,935.0 | 3,496.3 | 3,205.6 | 700.0 |
| Gravitar | 59.4 | 12.5 | 492.5 | 0.0 |
| Hero | 20,408.8 | 13,850.0 | 26,568.8 | 14,040.0 |
| IceHockey | -10.1 | -8.3 | -10.4 | -10.5 |
| Jamesbond | 700.0 | 431.3 | 264.6 | 500.0 |
| Kangaroo | 12,700.0 | 12,143.8 | 7,997.1 | 6,700.0 |
| Krull | 8,685.6 | 8,058.8 | 8,221.4 | 7,170.0 |
| KungFuMaster | 15,562.5 | 4,362.5 | 29,383.1 | 13,700.0 |
| NameThisGame | 9,056.9 | 7,241.9 | 6,548.8 | 3,700.0 |
| Phoenix | 5,295.6 | 4,326.9 | 3,932.5 | 1,880.0 |
| Pooyan | 2,859.1 | 1,677.2 | 4,000.0 | 330.0 |
| Qbert | 13,734.4 | 11,276.6 | 4,226.5 | 11,700.0 |
| Riverraid | 14,755.6 | 9,816.3 | 7,306.6 | 3,810.0 |
| RoadRunner | 54,568.8 | 49,118.8 | 25,233.0 | 50,900.0 |
| Robotank | 63.2 | 44.6 | 9.2 | 17.0 |
| Seaquest | 5,173.8 | 1,175.6 | 1,415.2 | 643.0 |
| TimePilot | 2,743.8 | 1,312.5 | -883.1 | 2,400.0 |
| UpNDown | 16,291.3 | 10,454.4 | 8,167.6 | 5,610.0 |
| VideoPinball | 1,007.7 | 1,140.8 | 85,351.0 | 0.0 |
| WizardOfWor | 187.5 | 443.8 | 975.9 | 500.0 |
| YarsRevenge | 28,897.9 | 20,738.9 | 18,889.5 | 19,505.4 |
| Zaxxon | 275.0 | 50.0 | -0.1 | 0.0 |

Table 2: Raw scores of 41 training Atari games for best performing multi-game models.