# OpenReview forum: "Multi-Game Decision Transformers"
_NeurIPS.cc/2022/Conference — NeurIPS 2022 Accept_

### Official Review · Reviewer_EPhF · 2022-07-10

**Rating:** 7
**Confidence:** 4
**Soundness:** 4 excellent
**Presentation:** 4 excellent
**Contribution:** 4 excellent

**Summary:**

This paper proposes training a single model on a lot of games, combining the RL approach with the same token prediction approach we've seen enable the recent very capable LLM results. The model they train does very well on the suite of Atari games and lets us ask questions like:

1. What is learned from playing all of those games?
2. Do the trends in vision and language hold for generalist RL agents?
3. Does this next-step tokenization prediction approach work for these games?
4. How do different online and offline methods work in this multi-game etup?
5. Does this idea of expert action inference, which comes from decomposing the next-step tokenization to explicitly model the expert prediction, improve upon BC?

The model is trained on trajectories collected in prior work and includes both expert and non-expert behavior. There are many baseline methods that they compare against. All in all, this is an extensive work which demonstrates that, while the model does not beat single-game performance (as expected), it is competitive across the majority of games.

**Questions:**

1. What happens if you do actually filter for only expert level trajectories? I get that you designed a policy that can skirt around this issue, but I think this would be good for understanding what happens when we use human experts as the behaviors to clone. That is largely our eventual purpose, so would be helpful to do that as a follow up. I suspect it wouldn't be that much extra work either on this paper ;).

2. Where is the Appendix? Some things are listed as being there, but I don't see it attached, nor is there a Supplemental... I might be being blind to this but other papers included their appendix explicitly in the paper.

**Limitations:**

The authors are honest about their limitations in the blurb at the end.

**Strengths And Weaknesses:**

Originality:

This paper is not particularly original. The only part of the algorithm that could be original is the Expert Action Inference inspired by Krause et al.

Clarity:

It's very clear. There's no issues here.

nit - L249: "We discuss these results *in* the Appendix".

Quality:

The quality is high as well. This paper sets out to do a goal - figure out how to train a single agent to a high level on a lot of games - and does it. It then supports that finding with relevant questions about whether it is capable of learning more (transfer), how it compares with online/offline methods, how it scales with size, and and how optimal vs suboptimal actions influence the learning. These all make sense as experiments to do and bolster the argument.

Significance:

The paper is significant in so far as it sets a bar for what comes next. It's somewhat of a successor (although not multi-modal) to One Model to Rule Them All, which previously was significant in showing that we can do a lot more in a single model than previously thought. It's not significant in being world-shaking. I think of this as being important in so far as a group showing the rest of the world that it's possible.

---

> ### Author Response · Authors · 2022-08-02
> **Author Response to Reviewer EPhF**
>
> Thank you for your thoughtful review! We are happy that the significance of this work is recognized.
>
> > Appendix
>
> We're not sure if there was a technical issue for you, but there was a “Supplementary Material:  ⬇ pdf” link on the OpenReview page (below the paper abstract), before we updated with the rebuttal revision. It seems that other reviewers were able to view the appendix, as they refer to it in their reviews. In case it's helpful, we have also updated the main paper to include the appendix instead of having a separate supplementary document. Please let us know if you are still not able to see it.
>
> > Filter for only expert level trajectories
>
> In the initial submission, we included comprehensive experiments (BC and DT) and discussions for “Learning Exclusively from Expert Data” in Appendix D in the supplementary. We hope that you are able to see that now in the revision.

---

### Official Review · Reviewer_wP61 · 2022-07-11

**Rating:** 7
**Confidence:** 3
**Soundness:** 3 good
**Presentation:** 3 good
**Contribution:** 3 good

**Summary:**

The paper empirically evaluates the ability of the recently proposed decision transformer model to generalize across Atari games. It also investigates other empirical questions (scaling behavior, transfer learning, baselines, etc.). The main empirical result is that the decision transformer model outperforms other generalist agents but loses to single-game agents. The paper demonstrates how an improved policy can be extracted using guided generation approaches similar to those used in language modeling and shows that scaling trends similar to those seen in vision and NLP hold for Atari games.

---------

UPDATE: I thank the authors for their detailed responses. After reading the other reviews and the responses, I've increased my score to "Accept".

**Questions:**

1. How does the test-time action selection work exactly? In particular, what happens if the P(expert|R) is very small? Can the full test-time procedure be included in the paper or appendix?

2. How sensitive are the empirical results to different seeds, hyper-params, etc?

**Limitations:**

The authors have described the main limitations. Based on the response to a question above, it might need to be expanded to include sensitivity wrt seeds, hyper-params, etc.

**Strengths And Weaknesses:**

Strengths
  + The paper investigates an important question (scaling behaviors of decision transformers) and conducts a detailed empirical investigation of the decision transformer in Atari games. The experiments are well constructed and nicely detailed.
  + The experimental results are novel (to my knowledge). The multi-game decision transformer seems to be the state-of-the-art in general game playing (although restricted to Atari games).
  + The paper is well written. It's easy to follow the main ideas and experiments. The paper also does a good job of placing its ideas into the larger body of work in sequence modeling, upside-down RL, transfer learning, etc.
  + There appears to be some algorithmic novelty in the guided action generation.

Weaknesses
  - As the paper mentions, it's unclear if these results hold outside of Atari games. However, evaluating this seems outside the scope of this paper.
  - I didn't fully understand the test-time action-selection described in Section 3.4. For example, what happens if P(expert|R) is very small? Full algorithmic pseudo-code might make it clearer.
  - It sounds like a single training run was used for each model. Is this correct? The answers in the checklist also mention error bars but I didn't see any in the figures. Also, the appendix mentions a single seed was used. Overall, it's unclear to me if these experiments are reproducible across seeds, hyper-params, etc. That said, given the large dataset and large computational cost, this may not be a major issue but perhaps this needs to be addressed somewhere in the paper.
  - Given that the main contributions of the paper are empirical, the above points make the overall impact of this paper somewhat limited. I'd be inclined to revise my opinion based on the authors response to the questions below.

---

> ### Author Response · Authors · 2022-08-02
> **Author Response to Reviewer wP61**
>
> Thank you for your feedback! Please let us know whether the following responses fully address your concerns or whether there are any points remaining, which we would be happy to address.
>
> > Test-time action selection
>
> In addition to the description in Section 3.4, we added a pseudo-code for expert action inference in Appendix A.3 in the rebuttal revision (note that we included the appendix in the main paper instead of having a separated supplementary document in the rebuttal revision). Hopefully the pseudo-code helps clarify implementation details. In addition, as we promised, we will release the actual python code.
>
> What we describe in Section 3.4 is adding a bias to adjust the categorical return distribution P(R|...) to make it easier to sample higher returns, by assigning P(expert|R) proportionally to return magnitude. The probability mass is shifted from lower return buckets, where P(expert|R) is small, to higher return buckets, where P(expert|R) is large. Very small P(expert|R) corresponds to very low return magnitude. Of course, we don’t want to make it easy to generate high returns that cannot be achieved from the current state. Therefore, it’s important to set a $\kappa$ to make P(expert|R) provide proper bias but not dominate.
>
> > Consistency across random seeds and hyperparameters
>
> For the 40M DT model, the mean and standard deviation of median scores across 5 random seeds are 0.79 and 0.06; the mean and standard deviation of IQM scores are 0.95 and 0.05. Compared to the differences to the baseline results and the results of other DT model sizes, we consider the variance to be relatively small.
>
> As for hyperparameters, although it would be difficult to perform sensitivity analysis for all of them, we note that we perform empirical hyperparameter and architecture searches on the smaller 40M model, and simply adopt the same configuration for the 200M model since it is too expensive to iterate on the 200M model. With the set of hyperparameters working the best for smaller models, it still manifests a clear scaling trend and transfer learning results. We hope this observation helps mitigate the concern of hyperparameter sensitivity.
>
> We added these discussions to Appendix A.1 in the rebuttal revision.

---

### Official Review · Reviewer_ZHCn · 2022-07-12

**Rating:** 8
**Confidence:** 4
**Soundness:** 3 good
**Presentation:** 4 excellent
**Contribution:** 3 good

**Summary:**

This paper introduces the possibility of having a generalist RL agent that is based on the transformer. The paper demonstrate the effectiveness of a single RL agent, and that its scaling trend follows those of language and vision. The idea is validated on Atari games.

**Questions:**

Is the result sensitive to the quantization scheme used in the paper?

**Ethics Review Area:**

["I don’t know"]

**Strengths And Weaknesses:**

Strengths:
The paper is easy to follow and the problem is well motivated, What really stands out the detailed description and the solid experiments performed. I think the results could have great impact on RL, and potentially making large transformer based offline RL training the new dominant paradigm for the field, especially with the scaling behavior.

Weakness
I guess the main weakness is the novelty since going from decision transformer to multitask decision transformer seems to be very straightforward. Nevertheless, I think this does not diminish the main value here.

---

> ### Author Response · Authors · 2022-08-02
> **Author Response to Reviewer ZHCn**
>
> Thank you for your thoughtful review. We are pleased that you found the description detailed and the experiments solid. We share your perspective that this approach has the potential to substantially change the field of RL.
>
> > Sensitivity to the choice of quantization scheme
>
> We did not consider other quantization schemes in our experiments, but we agree that is an interesting consideration for future work.

---

### Official Review · Reviewer_4PtS · 2022-07-17

**Rating:** 6
**Confidence:** 4
**Soundness:** 3 good
**Presentation:** 3 good
**Contribution:** 3 good

**Summary:**

This paper investigates the properties of decision transformer models trained on multiple Atari games using off-line RL. The model is found to be superior to multi-game models using other approaches, for example online non-transformer models or transformer models trained with behavioral cloning. The performance gap between single-game trained models vs. multi-game trained models, observed in some earlier works, was found in this study as well.

**Questions:**

Please see the limitations section below.

**Limitations:**

I understand that these are ultimately empirical results, but the authors could perhaps do a better job explaining why these observed results might make sense: for example, why online C51 doesn’t seem to do as well as the offline DT or why non-transformer models in general don’t seem to do as well as transformer models. The lack of any scaling with model size for CQL in Fig 5 (in fact inverse scaling with model size in Fig 5b!) also seems surprising a priori, for example. What is the hypothesized explanation for this?

In Figure 5a, the decision transformer is claimed to exhibit power law scaling with model size, but this doesn’t seem correct. First of all, it's not really possible to conclude a power law or anything like that from three data points only with any degree of confidence. Secondly, the figure shows a roughly linear increase in performance for the decision transformer in a semilogx plot. This corresponds to a logarithmic improvement in performance with model size (in fact it seems like the scaling on a linear-linear plot will be worse than logarithmic because the trend seems slightly slower than linear in the semilogx plot in Fig 5a). A power law would be linear in a log-log plot.

Also relatedly, were separate hyperparameter searches performed for different model sizes in the scaling experiments in Figure 5 and was the amount of compute (FLOPs) controlled between different model sizes (need to train smaller models for a larger number of iterations)? If not why not? If not, wouldn't that possibly underestimate the performance of the smaller size models in these experiments, as demonstrated, for example, in the new scaling laws paper by Hoffmann et al. (2022)?

In section 4.6, the best model rollouts are compared against the best training data. But, I’m really not sure if this is a fair comparison. With enough samples, one is essentially guaranteed to surpass the top training data eventually. Shouldn’t the model be expected to do better on average than the expert rollouts in the training data ideally? The performance gap between single-game and multi-game models shown in Fig. 1 (lack of positive transfer between games on average), also observed in other works, suggests this is not really the case yet with multi-game models.

**Strengths And Weaknesses:**

Strengths:

The experiments here are interesting and insightful, likely to be of general interest to the ML community. The work also seems technically sound.

Weaknesses:

Please see the limitations section below.

---

> ### Author Response · Authors · 2022-08-02
> **Author Response to Reviewer 4PtS (Part 2/2)**
>
> > On the gap between specialist agents and a generalist agents, and comparing the best rollouts from model and data
>
> We acknowledged that the performance gap still exists, as we noted in the third paragraph of the introduction: *“We are not striving for mastery or efficiency that game-specific agents can offer, as we believe we are still in early stages of this research agenda.”*. In this work, our focus is on making a significant step on learning high-performing generalist agents. As for comparing the best rollouts from model and data, “surpassing the top training data with enough data” is an interesting hypothesis to explore, but is largely outside the scope of this work. However, to potentially achieve such a result, we think the right method is still required. As we show in the paper, when using BC on the training data, the best evaluation samples were always worse than the best training examples, since its objective encourages learning the mean behavior instead of the optimal behavior. Other multi-game baselines such as CQL are also very far from achieving this goal. We think our empirical results are valuable, hopefully pointing to a bright future for the generalist model.

---

> ### Author Response · Authors · 2022-08-02
> **Author Response to Reviewer 4PtS (Part 1/2)**
>
> Thank you for your thoughtful feedback! Please let us know whether the following responses fully address your concerns or whether there are any points remaining, which we would be happy to address. We will incorporate our discussion into the post-rebuttal final revision.
>
> > Why online C51 doesn’t seem to do as well as the offline DT
>
> While it is difficult to definitively attribute causes to our observed empirical results, we believe the apparent advantage of offline DT compared to online multi-game methods like C51 may be explained due to classical differences between online and offline settings in RL [[Levine, et al 2020](https://arxiv.org/abs/2005.01643)]. Namely, online methods must balance exploration with the ability to learn and generalize from experience and this could be challenging in the multi-game setting, whereas offline DT only needs to learn to distill and generalize from the fixed multi-game experience given to it (collected by specialist DQN agents [[Agarwal et al, 2019](https://arxiv.org/abs/1907.04543)]). As we showed in the paper, the published IMPALA online agent performance is similar to our C51 agent, which is further evidence of this perspective. Beyond the difference between online and offline, one could also argue that C51 suffers from more training instability than DT due to the use of a temporal difference (TD) loss (see discussion of CQL two paragraphs below). We added these additional discussions into the appendix of the rebuttal revision (due the 9-page limit of rebuttal revision). Note that we included the appendix in the main paper instead of having a separated supplementary document in the rebuttal revision.
>
> > Why non-transformer models in general don’t seem to do as well as transformer models
>
> While the reasons for the benefits of transformer architecture over other neural networks are still an open question in general [see e.g., [Raghu, et al 2021](https://arxiv.org/abs/2108.08810)], our hypothesis is that transformers allow for easier discovery of correlations between components of the input and output, due to the fact that transformers process the input as a flat sequence with attention allowed between any patch, action, or return token. We added these into Appendix E in the rebuttal revision.
>
> > The lack of any scaling with model size for CQL in Fig 5 also seems surprising a priori
>
> In our experience, temporal difference (TD) methods suffer greater instability with larger model size in the multi-game setting, and this leads to the “inverse” scaling observed in our plots. This is one reason why we chose to evaluate CQL as a baseline (which is a combination of TD and behavioral cloning) – attempts at other objectives closer to pure TD (C51, DQN, DDQN) led to even worse results. We note that similar conclusions about instability with respect to network size have been made by other work [[Furuta, et al 2021](https://arxiv.org/abs/2103.17258)]. We added these into the appendix of the rebuttal revision.
>
> > Power law scaling
>
> We changed all "power law scaling" to just "scaling" in the rebuttal revision.
>
> > How hyperparameter searches were done
>
> Hyperparameter search is only done on the 40M model. As we noted in the paper, the 200M model takes 8-10 days to train and thus is too expensive to iterate on. We simply used the best configuration we found on the 40M model for the 200M model. We have clarified this in Appendix A.1 in the rebuttal revision.
>
> > Control the amount of compute (FLOPs)
>
> The performance of the 40M model does appear to flatten out. We have experimented with training it for 20M steps (doubled from the default 10M steps) and the performance does not change significantly. We hope that mitigates the concern. In the future, it would be interesting to see if the findings of [Hoffmann et al. (2022)](https://arxiv.org/abs/2203.15556) can be transferred to generalist agent learning where modality and data distribution are very different. [Hoffmann et al. (2022)](https://arxiv.org/abs/2203.15556) is concurrent with our work, so we did not consider searching for the optimal model size with a fixed FLOPs budget in the scope of this work.

---

### Meta-Review · Area_Chair_d2xZ · 2022-08-26

**Recommendation:** Accept
**Confidence:** Certain

**Metareview:**

This paper demonstrates the generalization abilities of decision transformers relative to other approaches in multi-task settings. The reviewers found the topic interesting and the results compelling, unanimously supporting its inclusion in the conference programme. Accept.

**Award:**

No

---

### Decision · Program_Chairs · 2022-09-14

Accept